# Risk Factors for Long-Term Delayed Gastric Emptying and Its Impact on the Quality of Life After Laparoscopic Pylorus-Preserving Gastrectomy in Patients with Gastric Cancer: Secondary Analysis of the Prospective Multicenter Trial KLASS-04

**DOI:** 10.3390/cancers17152527

**Published:** 2025-07-30

**Authors:** Young Shick Rhee, Sang Soo Eom, Bang Wool Eom, Dong-eun Lee, Sa-Hong Kim, Hyuk-Joon Lee, Young-Woo Kim, Han-Kwang Yang, Do Joong Park, Sang Uk Han, Hyung-Ho Kim, Woo Jin Hyung, Ji-Ho Park, Yun-Suhk Suh, Oh Kyoung Kwon, Wook Kim, Young-Kyu Park, Hong Man Yoon, Sang-Hoon Ahn, Seong-Ho Kong, Keun Won Ryu

**Affiliations:** 1Center for Gastric Cancer, National Cancer Center, Goyang 10408, Republic of Korea; 2Department of Surgery, Inje University College of Medicine, Ilsan Paik Hospital, Goyang 10380, Republic of Korea; 3Biostatistics Collaboration Team Research Institute, National Cancer Center, Goyang 10408, Republic of Korea; 4Department of Surgery, Seoul National University Hospital, Seoul 03080, Republic of Korea; 5Department of Surgery and Cancer Research Institute, Seoul National University College of Medicine, Seoul National University Hospital, Seoul 03080, Republic of Korea; 6Department of Surgery, Seoul National University College of Medicine, Seoul National University Bundang Hospital, Seongnam 13620, Republic of Korea; 7Department of Surgery, School of Medicine, Ajou University, Suwon 16499, Republic of Korea; 8Department of Surgery, Yonsei University Severance Hospital, Seoul 03722, Republic of Korea; 9Department of Surgery, Gyeongsang National Univeristy College of Medicine, Jinju 52727, Republic of Korea; 10Department of Gastrointestinal Surgery, Kyungpook National University Chilgok Hospital, Daegu 41404, Republic of Korea; 11Department of Surgery, Cheju Halla General Hospital, Jeju City 63127, Republic of Korea; 12Department of Surgery, Chonnam National University Medical School, Hwasun 58128, Republic of Korea

**Keywords:** pylorus-preserving gastrectomy, delayed gastric emptying, quality of life

## Abstract

Delayed gastric emptying (DGE) is a well-known complication of laparoscopic pylorus-preserving gastrectomy (LPPG). Patients who underwent LPPG in the KLASS-04 trial showed an unneglectable incidence—21/124 patients (16.3%)—of long-term DGE compared to those who underwent laparoscopic distal gastrectomy. This study aimed to identify the multifactorial risk factors associated with DGE and analyze the quality of life (QoL) of patients with DGE following LPPG. Patients without previous abdominal surgery had a higher incidence of DGE in the univariate (32% vs. 4.8%, *p* = 0.011) and logistic regression analyses (odds ratio: 0.106, 95% confidence interval: 0.014–0.824, *p* = 0.032). Patients with DGE reported more symptoms of nausea and vomiting (*p* = 0.004), constipation (*p* = 0.04), and a dry mouth (*p* = 0.005). No clinicopathological or surgical factors, other than the absence of a previous surgical history, were identified as multifactorial risk factors for DGE. However, DGE had a negative impact on the QoL of patients.

## 1. Introduction

Gastric cancer is the fifth most common cancer worldwide, and was the fourth most common cancer in Korea as of 2021, accounting for 10.6% of all cancers [1,2]. The national screening programs for gastric cancer are well organized in Korea and Japan because of the high incidence rate of gastric cancer in these countries; consequently, most gastric cancers are diagnosed at an early stage, with EGC accounting for 63.1% of all diagnoses in Korea in 2023 [3].

Since the survival rate of surgically treated early gastric cancer (EGC) is >90%, several surgical options are currently being investigated to improve the quality of life (QoL) of patients, which is impaired after standard gastrectomy, such as when postgastrectomy syndrome occurs [4]. Pylorus-preserving gastrectomy (PPG), initially devised for the treatment of peptic ulcer disease, is currently endorsed as an alternative surgical approach for clinical T1N0 EGC located in the middle third of the stomach, provided that a minimum distance of 4 to 5 cm is maintained above the pylorus. This recommendation is supported by the guidelines of the Korean Gastric Cancer Association, the Japanese Gastric Cancer Association, and the Chinese Society of Clinical Oncology, with the expectation of preserving the function of the pylorus. The KLASS-04 trial, a prospective multicenter randomized control trial (RCT) comparing the outcomes of laparoscopic PPG (LPPG) and laparoscopic distal gastrectomy (LDG) for cT1N0M0 gastric cancer in the mid-portion of the stomach, showed that LPPG has more favorable outcomes in terms of nutritional status, gallstone formation, and bile reflux compared to LDG, but has the pitfalls of delayed gastric emptying (DGE) and esophageal reflux [5].

DGE, a well-known drawback of PPG, can mostly be treated with diet modification and conservative medical treatment; however, it not only impairs the QoL of patients, but it also sometimes requires additional interventions for management. Several individual risk factors for DGE have been reported, such as damage to the hepatic branch of the vagus nerve (HBVN) [6], infrapyloric artery (IPA) [7], infrapyloric vein (IPV) [8], and short antral cuff [9]; however, these factors interact with one another, and cannot be attributed to a single factor. These well-known factors were considered in the study protocol, and the surgical procedure was well controlled to prevent DGE in the KLASS-04 trial.

In this study, the KLASS-04 trial data underwent secondary analysis to determine the multifactorial risk factors for DGE, evaluate its impact on patient QoL, and improve surgical outcomes after LPPG. Patients who underwent LPPG were categorized into DGE and non-DGE groups.

## 2. Materials and Methods

### 2.1. Study Design and Patients

The patient inclusion criteria for the KLASS-04 trial were the following: (1) an age between 20 and 80 years, with an Eastern Cooperative Oncology Group (ECOG) performance status of 0 or 1; (2) histologically confirmed gastric adenocarcinoma; (3) a preoperative diagnosis of cT1N0M0 using gastroscopy, endoscopic ultrasound, and abdominal computed tomography; and (4) a tumor proximal/distal margin of at least 5 cm from the gastroesophageal junction/pylorus.

Patients with a history of other cancers, synchronous EGC or adenoma of the antrum, previous gastric surgery—including gastrojejunostomy—or the presence of other malignancies within the previous 5 years (except for cured basal cell carcinoma or in situ cervical cancer) were excluded.

Surgeon qualification was determined based on the following criteria: (1) a minimum of 50 surgeries performed for both LDG and open distal gastrectomy; (2) experience with at least five cases of LPPG; and (3) affiliation with an institution performing a minimum of 80 gastrectomy procedures annually.

The surgical procedure of LPPG was standardized as detailed below.

Dissection of the #6 lymph nodes was carried out meticulously to preserve the infrapyloric vessels. Dissection of the #5 lymph nodes was omitted, and the right gastric artery arcade was ligated at a point 3 cm proximal to the pylorus. Gastro-gastrostomy was performed using an extracorporeal hand-sewn anastomosis technique.

In accordance with the KLASS-04 criteria, DGE was defined as “nearly normal diet residue” at least once in the endoscopic follow-up at 1, 2, and 3 years after the surgery [5].

The primary hypothesis was that there would be differences in the clinicopathological features and surgical outcomes between the two groups. The secondary hypothesis was that the QoL would be worse in the DGE group than in the non-DGE group.

The clinicopathological features analyzed included age, sex, body mass index (BMI), ECOG, diabetes mellitus (DM), history of a previous abdominal operation, tumor location, tumor size, pathological stage and final stage, lymph node (LN) dissection level, and surgical outcomes—including the operation time, intraoperative bleeding amount, IPA type and injury, HBVN injury, length of the remaining antral cuff, distance of the distal margin to the pylorus, proximal and distal margin of the tumor, preservation of the celiac branch of the vagus nerve, number of resected #6 LNs and total LNs, and overall morbidity.

QoL analysis was performed using the EORTC C30 and STO22 questionnaire surveys, which were administered preoperatively and at 6 months, 1, 2, and 3 years after the surgery.

### 2.2. Statistical Analysis

To compare the distribution based on the presence of DGE, the Chi-square test or Fisher’s exact test was performed for categorical variables, and the Wilcoxon rank sum test was used to analyze continuous variables. Logistic regression analysis was performed to identify the factors affecting DGE and to estimate the odds ratios (ORs) with 95% confidence intervals (CIs). Generalized linear mixed models were used for the QoL analysis. The primary index of the QoL analysis was the group *p*-value rather than the group × time interaction *p*-value, since the primary objective was to evaluate the differences in QoL scores between the groups across all time points, rather than differences in temporal trends. The analysis was conducted by comparing the changes in QoL at each time point with the baseline. Statistical significance was set at *p* < 0.05, and statistical analyses were performed using SAS version 9.4 (SAS Institute Inc., Cary, NC, USA) and R version 4.3.3 (R Foundation for Statistical Computing, Vienna, Austria).

## 3. Results

### 3.1. Patient Recruitment

A total of 283 patients were enrolled in the KLASS-04 trial. After excluding 27 screening failures, 256 patients were randomized to the LDG or LPPG arm. After excluding 129 patients who underwent LDG and three patients who underwent laparoscopic total gastrectomy, 124 patients who underwent LPPG were included in the secondary analysis. Among them, 21 were diagnosed with DGE, while 103 exhibited no endoscopic findings indicative of DGE (Figure 1).

### 3.2. Clinicopathological Features

Patients without a history of surgery showed a higher incidence of DGE (32% vs. 4.8%, *p* = 0.013; Table 1). The BMI and presence of underlying DM did not differ significantly between the two groups (23.5 vs. 23.6, *p* = 0.8964 and 8.7% vs. 4.8%, *p* = 0.468, respectively). No significant differences were observed in either the pT classification or the pathological stage of the disease (*p* = 0.853, *p* = 0.504).

### 3.3. Surgical Outcomes

No significant differences in operative details were found between the two groups (Table 2).

The previously known risk factors for DGE after LPPG, namely injury to the HBVN, IPA, and IPV and antral cuff length, showed no difference between the two groups (2% vs. 4.8%, *p* > 0.99; 1% vs. 0%, *p* > 0.99; 3.9% vs. 0%, *p* > 0.99; 4 vs. 4, *p* = 0.9194, respectively). The number of resected LNs, including #6 LNs, #9 LNs, and total LNs, also showed no significant differences (6 vs. 5, *p* = 0.526; 2 vs. 2.5, *p* = 0.471; 49 vs. 39, *p* = 0.113). Intraoperative bleeding did not differ significantly between the groups (31.5 mL vs. 50 mL, *p* = 0.2475). The mean operation time demonstrated a nearly significant difference, with the DGE group exhibiting shorter operation times compared to the non-DGE group (190 min vs. 199 min, *p* = 0.08).

### 3.4. Logistic Regression Analysis

Univariate logistic regression was performed to determine whether clinicopathological characteristics and surgical factors affected DGE (Table 3). Patients with a history of previous abdominal surgery showed a lower tendency for DGE in both the univariate and multivariate analyses (OR: 0.106, 95% CI: 0.014–0.824, *p* = 0.032; OR: 0.094, 95% CI: 0.012–0.757, *p* = 0.026, respectively). The operative time and number of resected LNs showed marginal tendencies toward DGE in the univariate analysis. The operative time was slightly shorter in the DGE group than in the non-DGE group, while the number of resected LNs was slightly higher (OR: 0.989, 95% CI: 0.977–1.002, *p* = 0.093; OR: 1.029, 95% CI: 0.997–1.061, *p* = 0.074, respectively).

### 3.5. QoL

In the EORTC C30, patients with DGE exhibited more nausea and vomiting (*p* = 0.004) and constipation (*p* = 0.04). Insomnia exhibited nearly-significant differences between the groups (*p* = 0.099), and the time × group interaction analysis indicated a trend toward worsening insomnia in the DGE group compared with the non-DGE group (*p* = 0.03). There were no significant differences in the other scales (Figure 2).

In the EORTC STO22, aside from dry mouth (*p* = 0.005), no significant differences were observed between the two groups. While the eating restriction score demonstrated a marginally significant difference between the groups (*p* = 0.096), no significant differences were noted at the 3-year follow-up (Figure 3).

## 4. Discussion

The design of the LPPG surgical protocol considered previously identified individual risk factors, including injury to the HBVN, IPA, and IPV and a short antral cuff length, as well as the qualifications of the operating surgeon. Despite these considerations, the incidence of DGE was significantly higher in the LPPG cohort than in the LDG group, with rates of 16.3% and 3.9%, respectively (*p* = 0.001). This study was performed to identify the multifactorial risk factors for DGE and analyze the impact of DGE on the QoL of patients who underwent LPPG in the KLASS-04 trial. Although novel risk factors were not identified, except for a previous history of abdominal surgery, this study demonstrates the negative effect of DGE on patient QoL. This is the first comprehensive analysis using data from a prospective RCT of DGE after LPPG.

Because patients with prior gastric surgeries were excluded from the study, the history of previous abdominal surgeries primarily involved colonic or gynecological procedures, which likely had a minimal direct impact on LPPG. None of the patients underwent pancreatic or hepatic surgery. The absence of previous surgeries, along with shorter operation times, was associated with a higher incidence of DGE with marginal statistical significance in the univariate analysis, suggesting that less radical surgical interventions may be correlated with long-term DGE outcomes.

Despite high BMI being implicated as a risk factor for DGE [10], our analysis revealed no significant difference in BMI between the DGE and non-DGE groups, possibly attributable to the lower prevalence of overweight individuals in the Korean population. Furthermore, DM, which is recognized as a risk factor for general gastroparesis [11], did not increase the risk of DGE after LPPG. The lack of significant differences in previously established individual risk factors between patients who did and did not develop DGE suggests that adherence to the initial surgical protocol was maintained throughout the study. Future investigations are warranted to elucidate the pathophysiological mechanisms underlying DGE following LPPG to improve surgical outcomes.

In the QoL analysis, patients with DGE experienced more nausea and vomiting and constipation. Given the relatively young average age of patients undergoing LPPG (55.7 years), the risk of aspiration due to vomiting was lower; however, the high incidence of DGE associated with LPPG, which is correlated with increased vomiting, makes this procedure less advisable for older patients.

In the KLASS-04 study, only 2 out of the 21 patients with endoscopic evidence of food retention were clinically diagnosed with DGE at the same time. Furthermore, 14 out of the 16 patients that were clinically diagnosed with DGE did not exhibit “nearly normal food residue” on concurrently performed endoscopy. This discrepancy is consistent with the results of previous studies, wherein the presence of residual food during endoscopy does not reliably correlate with clinical DGE [12]. Nevertheless, this does not negate the utility of endoscopy as a diagnostic modality for DGE, as patients with endoscopically defined DGE experience several statistically significant or near-significant DGE-related symptoms, including nausea and vomiting, constipation [13], and eating restrictions, as evidenced by the QoL analysis.

Among the 21 patients who demonstrated endoscopic evidence of food retention, 17 cases were observed in the first year of follow-up, 10 cases in the second year, and only one case was noted in the third years. This decreasing trend is consistent with previous studies [14], suggesting that DGE typically improves over time.

Our study has several limitations. First, DGE was not defined based on clinical symptoms corroborated by gastric scintigraphy, but was rather defined through endoscopic findings. Gastric scintigraphy requires several hours to perform and precise timing, which poses challenges in an outpatient setting. Therefore, endoscopy—a routine follow-up examination after gastric cancer surgery—was utilized as the diagnostic tool. Second, the surgical protocol for LPPG in the KLASS-04 trial involved extracorporeal anastomosis, following earlier studies reporting higher rates of DGE with intracorporeal anastomosis [15]. However, more recent evidence suggests that intracorporeal anastomosis demonstrates outcomes comparable to those obtained with extracorporeal techniques [16], and it is increasingly being adopted as a standard practice. Third, the sample size of 124 patients was insufficient to achieve a statistical power of 80% for this analysis. This limitation may have contributed to the negative findings of the study. Therefore, further research with a larger cohort is warranted to validate these results.

## 5. Conclusions

Despite the strict surgical protocol and procedure, considering well-known individual risk factors for DGE and surgeon qualifications, the LPPG group of the KLASS-04 trial exhibited a considerable incidence of DGE. No clinicopathological or surgical factors, other than the absence of a previous surgical history, were identified as multifactorial risk factors for DGE; however, given the negative impact of DGE on patient QoL, further research is necessary to develop strategies to improve outcomes in DGE following LPPG.

## Figures and Tables

**Figure 1 cancers-17-02527-f001:**
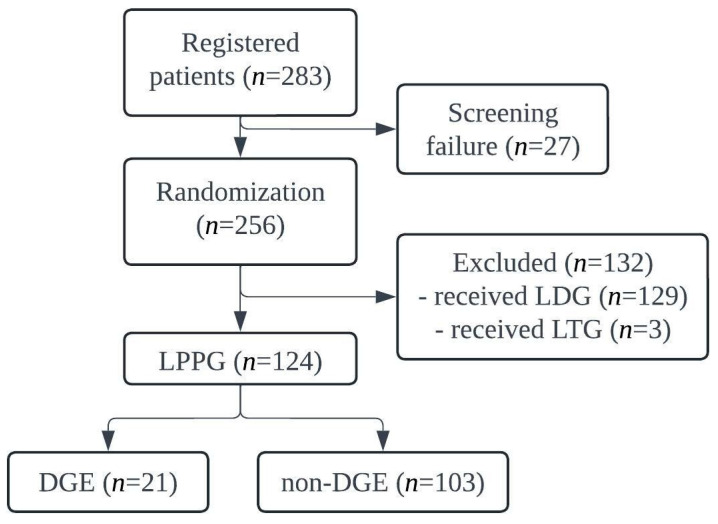
Consort diagram. LDG, laparoscopic distal gastrectomy; LTG, laparoscopic total gastrectomy; LPPG, laparoscopic pylorus-preserving gastrectomy; DGE, delayed gastric emptying.

**Figure 2 cancers-17-02527-f002:**
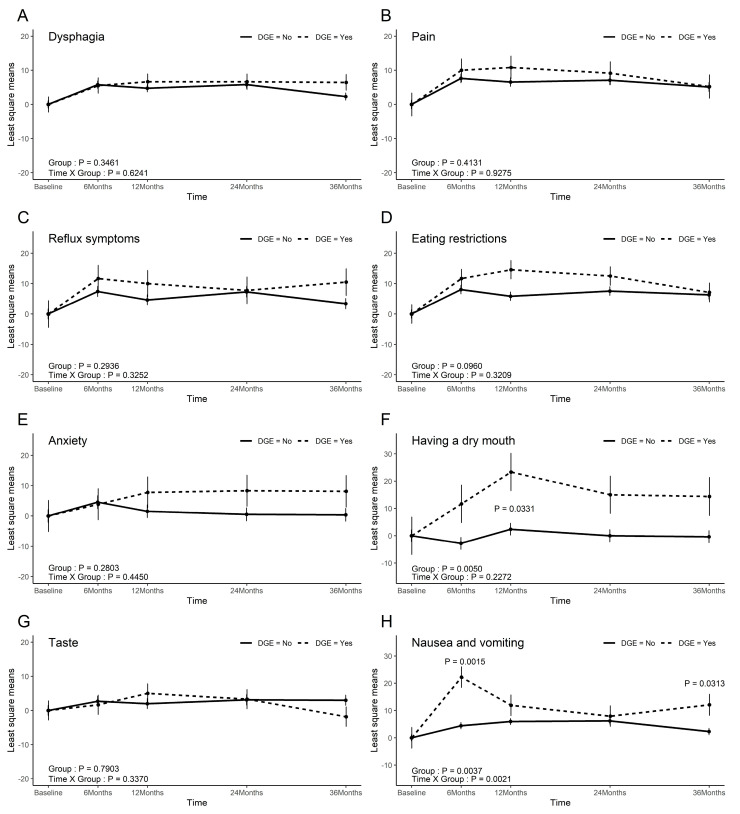
EORTC-C30 scores. (**A**) Global health status; (**B**) Physical functioning; (**C**) Role functioning; (**D**) Emotional functioning; (**E**) Cognitive functioning; (**F**) Social functioning; (**G**) Fatigue; (**H**) Nausea and Vomiting; (**I**) Pain; (**J**) Dyspnea; (**K**) Insomnia; (**L**) Appetite loss; (**M**) Constipation; (**N**) Diarrhea; (**O**) Financial difficulites.

**Figure 3 cancers-17-02527-f003:**
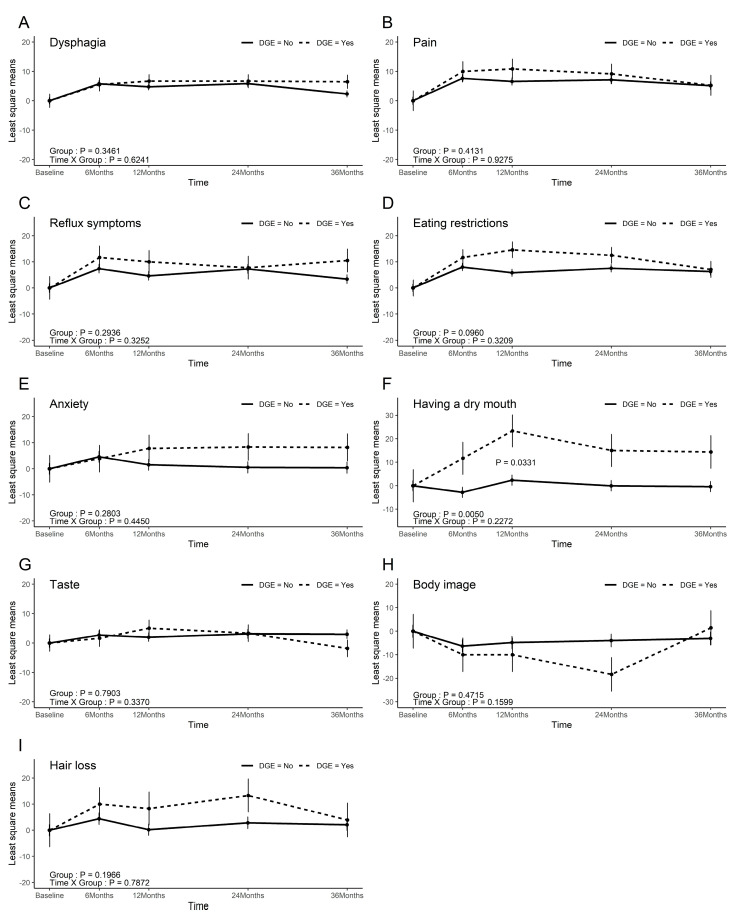
EORTC-STO22 scores. (**A**) Dysphagia; (**B**) Pain; (**C**) Reflux symptoms; (**D**) Eating restrictions; (**E**) Anxiety; (**F**) Having a dry mouth; (**G**) Taste; (**H**) Body image; (**I**) Hair loss.

**Table 1 cancers-17-02527-t001:** The clinicopathological features of the patients (*n* = 124).

	Delayed Gastric Emptying		*p*-Value
	No	No	
	*n* = 103	*n* = 103	
Age (years), mean ± standard deviation	55.2 ± 10.1	57.8 ± 13.3	0.31 *
Sex ratio (M:F)	50:53:00	8:13	0.382
BMI, mean ± standard deviation	23.5 ± 2.7	23.6 ± 2.4	0.896 *
ECOG			
0	100	19	0.199
1	3	2	
DM			
No	94	20	>0.99
Yes	9	1	
Previous surgical history			
No	70	20	0.011
Yes	33	1	
Tumor location			
Upper	1	0	0.376
Middle	80	14	
Low	20	7	
Tumor size (mm), median	20 (0–75)	16 (0.8–55)	0.05 ^#^
(Min–Max)			
pT classification			
T1a	62	13	0.853
T1b	35	7	
T2	3	1	
T3	1	0	
pN classification			
N0	91	19	>0.99
N+	11	2	
LN dissection level			
D1	0	1	0.169
D1+	103	20	
Pathologic stage			
IA	88	18	0.504
IB	12	2	
IIA	1	1	
IIB	1	0	

Values are *n* unless otherwise indicated. χ^2^ test or Fisher’s exact test, except for * *t*-test and ^#^ Wilcoxon rank sum test. BMI, body mass index; ECOG, Eastern Cooperative Oncology Group; DM, diabetes mellitus; LN, lymph node.

**Table 2 cancers-17-02527-t002:** The surgical outcomes of the enrolled patients (*n* = 124).

	Delayed Gastric Emptying		*p*-Value
	No	Yes	
	(*n* = 103)	(*n* = 21)	
Operation time (min) *	199 (105–275)	190 (130–243)	0.08
Blood loss (cc) *	31.5 (5–1000)	50 (5–220)	0.248
IPA type			
ASPDA (distal)	26	4	0.944 ^‡^
RGEA (caudal)	39	9	
GDA (proximal)	34	7	
None or unknown	4	1	
IPA injury			
No	101	21	>0.99 ^‡^
Yes	1	0	
IPV injury			
No	98	21	>0.99 ^‡^
Yes	4	0	
HBVN injury			
No	100	20	>0.99 ^‡^
Yes	2	1	
Length of antral cuff *	4 (3–8.6)	4 (3–6)	0.919
Proximal margin *	2.2 (0.2–13.5)	2.2 (1.1–5)	0.954
Distal margin *	2.7 (0.2–13.8)	3.6 (0.2–11)	0.427
Preservation of the CBVN			
No	78	14	0.346 ^†^
Yes	24	7	
Resected LN 6 *	6 (0–20)	5 (1–14)	0.526
Resected LN 9 *	2 (0–11)	2.5 (0–5)	0.471
Resected total LN *	34 (16–88)	39 (15–82)	0.113
Morbidity			
No	84	15	0.227 ^‡^
Yes	17	6	

Values are *n* except for * median (min–max). Wilcoxon rank sum test, except for ^†^ χ^2^ test and ^‡^ Fisher’s exact test. IPA, infrapyloric artery; ASPDA, anterior superior pancreaticoduodenal artery; RGEA, right gastroepiploic artery; GDA, gastroduodenal artery; IPV, infrapyloric vein; HBVN, hepatic branch of vagus nerve; CBVN, celiac branch of vagus nerve; LN, lymph node.

**Table 3 cancers-17-02527-t003:** Logistic regression analysis (*n* = 124).

	N	DGE	Univariate Analysis		Multivariate Analysis	
			OR (95% CI)	*p*-Value	OR (95% CI)	*p*-Value
Age	124	21	1.024 (0.979–1.07)	0.309		
Sex						
Female	66	13	1 (ref)			
Male	58	8	0.652 (0.249–1.707)	0.384		
BMI	124	21	1.012 (0.845–1.213)	0.895		
ECOG						
0	119	19	1 (ref)			
1	5	2	3.509 (0.549–22.432)	0.185		
DM						
No	114	20	1 (ref)			
Yes	10	1	0.522 (0.063–4.358)	0.548		
Previous surgical history						
No	90	20	1 (ref)		1 (ref)	
Yes	34	1	0.106 (0.014–0.824)	0.032	0.094 (0.012–0.757)	0.0263
Tumor location						
Low	27	7	1 (ref)	−0.39		
Middle	94	14	0.492 (0.177–1.366)	0.584		
Upper	1	0	0.913 (0.009–91.151)	0.91		
Tumor size	123	21	0.973 (0.935–1.012)	0.165		
pT classification						
T1a	75	13	1 (ref)	−0.936		
T1b	42	7	0.978 (0.362–2.640)	0.671		
T2	4	1	1.984 (0.216–18.250)	0.687		
T3	1	0	1.573 (0.017–147.919)	0.921		
pN classification						
N0	110	19	1 (ref)	−0.722		
N1	10	1	0.741 (0.114–4.815)	0.45		
N2	2	1	4.694 (0.281–78.349)	0.364		
N3b	1	0	1.615 (0.018–147.941)	0.979		
Pathologic stage						
IA	106	18	1 (ref)	−0.743		
IB	14	2	0.957 (0.215–4.258)	0.526		
IIA	2	1	4.785 (0.286–80.046)	0.385		
IIB	1	0	1.639 (0.018–151.093)	0.996		
Operation time (min)	124	21	0.989 (0.977–1.002)	0.093	0.989 (0.976–1.002)	0.1002
Blood loss (cc)	124	21	1 (0.996–1.004)	0.919		
IPA type						
ASPDA (distal)	30	4	1 (ref)	−0.936		
RGEA (caudal)	48	9	1.5 (0.418–5.383)	0.798		
GDA (proximal)	41	7	1.338 (0.354–5.06)	0.992		
No or unknown	5	1	1.625 (0.143–18.473)	0.825		
IPA injury						
No	122	21	1 (ref)			
Yes	1	0	1.577 (0.017–148.94)	0.844		
IPV injury						
No	119	21	1 (ref)			
Yes	4	0	0.509 (0.019–13.8)	0.689		
HBVN injury						
No	120	20	1 (ref)			
Yes	3	1	2.5 (0.216–28.913)	0.463		
Length of antral cuff	123	21	0.911 (0.516–1.609)	0.748		
Proximal margin	122	21	0.951 (0.728–1.243)	0.716		
Distal margin	122	21	1.079 (0.913–1.274)	0.373		
CBVN preservation						
No	92	14	1 (ref)			
Yes	31	7	1.625 (0.588–4.489)	0.349		
Resected LN 6	123	21	0.96 (0.847–1.089)	0.528		
Resected LN 9	121	20	0.895 (0.719–1.114)	0.322		
Resected total LN	123	21	1.029 (0.997–1.061)	0.074	1.031 (0.997–1.065)	0.0726
Morbidity						
No	99	15	1 (ref)			
Yes	23	6	1.976 (0.671–5.824)	0.217		

Values are *n*. BMI, body mass index; ECOG, Eastern Cooperative Oncology Group; DM, diabetes mellitus; IPA, infrapyloric artery; ASPDA, anterior superior pancreaticoduodenal artery; RGEA, right gastroepiploic artery; GDA, gastroduodenal artery; IPV, infrapyloric vein; HBVN, hepatic branch of vagus nerve; CBVN, celiac branch of vagus nerve; LN, lymph node.

## Data Availability

The datasets generated and/or analyzed during the current study are not publicly available due to the expiration of the original study data (KLASS-04), but they can be obtained from the corresponding author upon reasonable request.

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
