# Peer review of "Risk Factors for Long-Term Delayed Gastric Emptying and Its Impact on the Quality of Life After Laparoscopic Pylorus-Preserving Gastrectomy in Patients with Gastric Cancer: Secondary Analysis of the Prospective Multicenter Trial KLASS-04"

_cancers, 2025, doi:10.3390/cancers17152527_

Round 1

Reviewer 1 Report

Comments and Suggestions for Authors

The authors present and interesting subgroup analysis of the KLASS 04 trial. This analysis focuses specifically on the incidence and clinical manifestation of DGE after Laparoscopic assisted Pylorus preserving gastrectomy for early gastric cancer. It brings novel information putting numbers to the theoretical concerns after pyloric preservation of DGE.  

The paper is well written and interesting however its applicability is limited to high volume centers seen large groups of patients with early gastric cancers located in the mid body of the stomach. The numbers are low and that may impact the analysis. Especially as the finding of more DGE associated with shorter surgeries in patients having no prior surgery is likely spurious and a statistical red herring more than a true finding and the authors need to limit the scope of their attribution of risk factors as with only 124 patients truly prove or disprove risk factors for a <20% complications is very difficult

Additionally, the authors should comment in the fact that some of their findings could be attributed to the handsewn extracorporeal anastomotic technique used more than to just the pyloric preservation.

Finally weight alterations related with DGE need to be added to the analysis as it is not clear if symptomatic versus asymptomatic DGE is a meaningful problem to have after surgery.  

Author Response

Comments 1: The numbers are low and that may impact the analysis. Especially as the finding of more DGE associated with shorter surgeries in patients having no prior surgery is likely spurious and a statistical red herring more than a true finding and the authors need to limit the scope of their attribution of risk factors as with only 124 patients truly prove or disprove risk factors for a <20% complications is very difficult.

We agree that the incidence of DGE in LPPG was not sufficient for the analysis, since the design of original study was calculated by comparing LDG and LPPG in respect of dumping syndrome, while this article included only LPPG patients in analysis.

Although we used fisher’s exact test and mann whitney U test to correct the sample size error, it is still not enough to secure power. Lacking of enough number of patient may affect power, which means false negative result, which is also very a shame to us, considering the negative result of the article. However, it was also inborn drawback of secondary analysis, and we suggest further research. We added comment about this in discussion section and highlighted in yellow.

Comments 2: The authors should comment in the fact that some of their findings could be attributed to the handsewn extracorporeal anastomotic technique used more than to just the pyloric preservation.

Thank you for your comment. Several articles previously proved that extracorporeal anastomosis has more complication than intracorporeal anastomosis. However, all the patients who received PPG in this article had extracorporeal anastomosis, so between DGE-group and non-DGE group, whether they received extracorporeal or intracorporeal anastomosis is not a matter of consideration.

The fact that 16.3% incidence of DGE in PPG group might be contributed by extracorporeal anastomosis, but that incidence rate itself is not included in the analysis of this article.

Comments 3: Weight alterations related with DGE need to be added to the analysis as it is not clear if symptomatic versus asymptomatic DGE is a meaningful problem to have after surgery.  

It must have been much better if we could relate it to weight loss as you commented. However, weight information on each visit was not acquired in original dataset, and due to the character of secondary analysis of multicenter study, it was impossible to achieve that information. Thank you.

Reviewer 2 Report

Comments and Suggestions for Authors

Thank you for the opportunity to review this important manuscript. Here are my comments and suggestions.

Line 27: Define LDG

Please describe the surgical technique for PPLG. The type of reconstruction is important. Then I can additionally review the manuscript.

Explain the pathophysiologic difference of anxiety and hair loss between groups.

Where are the data for the comparison between LDG and LPPG.

Author Response

Thank you for your comment

Line 27: Define LDG

We corrected the misuse of abbreviation.

Please describe the surgical technique for PPLG. The type of reconstruction is important. Then I can additionally review the manuscript.

Surgical procedure was added in study design section and highlighted as green.

Explain the pathophysiologic difference of anxiety and hair loss between groups.

Anxiety and hair loss showed no statistically significant difference between the groups.

Where are the data for the comparison between LDG and LPPG.

Our study focused on comparison of patients with or without DGE within LPPG, and patients who underwent LDG were not included in the study.

You can refer to original article of KLASS-04 to check data for the comparison between LDG and LPPG.

Reviewer 3 Report

Comments and Suggestions for Authors

Thank you for the opportunity to review this study investigating factors associated with DGE after laparoscopic pylorus preserving gastrectomy. It is a sub-analysis of the landmark KLASS-04 randomized clinical trial. Overall it is well-written and the methods are accurate. However, the study design failed to show any significant conclusions, limiting in part its value.

  1. A lot of comparisons lead to marginally non-significant results. Perhaps you should acknowledge the limited sample size in the DGE group as a limitation.
  2. What is the exact definition of delayed gastric emptying? You mention that the time point for evaluation was in 1,2 and 3 years. However, the definition of DGE usually states delaying gastric emptying for more than one week.
  3. How do you explain the fact that previous surgeries were associated with no DGE? I would expect the opposite finding, since a previous surgery leads to adhesions and potentially more copious mobilization of the stomach.

Author Response

Thank you for your comment.

1.A lot of comparisons lead to marginally non-significant results. Perhaps you should acknowledge the limited sample size in the DGE group as a limitation.

It is also a shame for us, too. However, it was also inborn drawback of secondary analysis, and we suggest further research. We added comment about this in discussion section and highlighted in yellow.

2.What is the exact definition of delayed gastric emptying? You mention that the time point for evaluation was in 1,2 and 3 years. However, the definition of DGE usually states delaying gastric emptying for more than one week.

Exact definition of DGE is ‘delayed emptying of food material in stomach without mechanical obstruction, proven by objective examinations such as scintigraphy’. And in clinical field, patients’ subjective symptom such as bloating, food intake limitation, nausea/vomiting is considered more. However, such subjective symptoms are difficult to be obtained without exception in multicenter outpatient follow up, since primary goal of the original study was about dumping syndrome. And also, scintigraphy is difficult to be performed in outpatient setting.

So we used endoscopic food retention to define DGE. And the fact that endoscopically defined DGE has correlation with patient QoL is also meaningful, since, as in our study, endoscopy is the most feasible way to screen DGE after gastric cancer surgery.

3.How do you explain the fact that previous surgeries were associated with no DGE? I would expect the opposite finding, since a previous surgery leads to adhesions and potentially more copious mobilization of the stomach.

As mentioned in the discussion section, most abdominal surgery history was of lower abdomen surgery such as colorectal or gynecological. So its relationship with surgery itself or postoperative delayed gastric emptying is difficult to declare. In one study, gastropexy after right colectomy decreased delayed gastric emptying. So adhesion from previous surgery and LPPG might somehow fix stomach and have preventative effect from delayed gastric emptying, but less likely.

Round 2

Reviewer 2 Report

Comments and Suggestions for Authors

Were there any differences in some serum laboratory parameters, for example, Hemoglobin, Fe, TIBC?

Could the pylorus-preserving procedure have worse oncologic outcomes?

Author Response

1. Were there any differences in some serum laboratory parameters, for example, Hemoglobin, Fe, TIBC?

LPPG has significantly better outcomes than LDG in terms of Hb, serum protein level, and abdominal fat loss. However, permission to access the raw data is expired, so it is now hard to evaluate whether there were difference in lab values in DGE group and non-DGE group of LPPG patients.

2. Could the pylorus-preserving procedure have worse oncologic outcomes?

According to 'Laparoscopic pylorus preserving gastrectomy vs distal gastrectomy for early gastric cancer; A multicenter randomized controlled trial (KLASS-04)' by HJ Lee et al., oncological outcomes are not statistically different. But since oncological outcomes in early gastric cancer are very good, it might be difficult to show statistical difference. 

Thank you for your comment